# High-Temperature Tribological Behavior of HDPE Composites Reinforced by Short Carbon Fiber under Water-Lubricated Conditions

**DOI:** 10.3390/ma15134508

**Published:** 2022-06-27

**Authors:** Wen Zhong, Siqiang Chen, Zhe Tong

**Affiliations:** 1The Key Laboratory of Fluid and Power Machinery, Ministry of Education, Xihua University, Chengdu 610039, China; hujj0627@126.com; 2Luzhou Laojiao Group Co., Ltd., Luzhou 646000, China; 3School of Mechanical Engineering, North University of China, Taiyuan 030051, China; zhetong@nuc.edu.cn

**Keywords:** HDPE composites, SCFs, water-lubrication, high-temperature

## Abstract

The polymer water-lubricated bearing is widely used in marine transmission systems, and the tribological properties can be improved by addition of inorganic nano-fillers. The aim of this study is to investigate the effect of SCFs and temperature on the water-lubricating properties of high-density polyethylene (HDPE) composites. HDPE composites reinforced by varying content of short carbon fibers (SCFs) were fabricated via twin-screw extrusion and injection molding techniques to study the hardness and surface wettability of those composites. The tribological properties under water-lubricated conditions were investigated through a pin-on-disk reciprocating tribometer under different temperatures. The results showed that the increase in hardness of HDPE composites reached maximum to 42.9% after adding 25 wt % SCFs. The contact angle also increased with the increase in SCFs content and reached a maximum of 95.2° as the amount of SCFs increased to 20 wt %. The incorporation of SCFs increased the wear resistance and lubricating property of HDPE composites at different temperatures. The HDPE composite containing 20 wt % SCFs showed the lowest friction coefficient of 0.076 at 40 °C, and the wear track depth reached a maximum of 36.3 mm at 60 °C. Based on the surface wetting property and wear analysis, potential effect mechanisms of fillers and temperature were discussed. The knowledge from this study is useful for designing the anti-wear water-lubricated polymer bearing.

## 1. Introduction

Compared to metal material, polymer matrix materials are increasingly used in aerospace, marine, and subsea equipment to reduce the weight and improve the corrosion resistance of key components [1,2]. Polymer matrix materials are the potential materials for a water-lubricated bearing due to its low friction coefficient and good fatigue resistance [3,4].

However, the disadvantage of the mechanical properties and wear resistance hinders its further application. The solid particles, such as layered particles and hard particles, were added to polymer matrix composites, which have greatly improved the tribological and mechanical properties, as well as wettability [5,6,7,8]. With high modulus and strength, carbon fiber and glass fiber can significantly enhance the mechanical properties and wear resistance of polymer matrix composites when used as additives [9,10]. The tribological and mechanical properties of polyetherimide (PEI) composites were improved after the addition of short carbon fiber and expanded graphite; the excellent bearing capacity of short carbon fiber and self-lubricating characteristics are considered to be the main reason [11]. It was found that the friction and wear modification of polyimide (PI) composites had been achieved by introducing multiscale CFs, the main mechanism is the decrease in stress concentration between the matrix and reinforcement phase caused by multiscale characteristics of fillers [12]. CNTs with high elastic modulus and strength are also usually used as additives to enhance the tribological and mechanical characters of polymers matrix composites, and the lubrication and wear mechanism is similar to SCFs reinforced composites [13,14,15].

The friction heat accumulation between lubricated contacts has greatly limited the application of polymer materials under dry sliding conditions, and water-lubrication can well improve this situation. the polymer materials possess high corrosion resistance compared to traditional metal materials and relatively low wear rate in water-lubricated environments; these factors make the polymer materials widely applied as a seal and bearing in water [16,17,18]. The different types of additives were also used to promote the water lubrication capacity and wear resistance of polymer materials, such as rubber, polyether-ether-ketone (PEEK), and phenolic [19,20,21]. It is reported that the addition of microcapsules can cause the decrease in the friction coefficient and wear loss of ultra-high molecular weight polyethylene (UHMWPE).The palmityl palmitate was released and attached to the surface of the counterpart during the sliding process, preventing the direct contact between composites and counterpart [4,22]. Similar to the dry sliding condition, the use of nano-diamond, CNTs, and graphene oxide (GO) can also improve the hardness, elasticity modulus, and transfer film forming property of polymer materials that results in a high wear resistance under water lubrication [23,24]. The surface wettability has a significant effect on lubrication film characteristics and water absorption of polymer materials [25,26], and is an important influence factor on tribological properties of polymer materials besides the mechanical properties. It is also found that the contact angle of polypropylene increases with the addition of reduced graphene oxide (rGO). The analysis reveals that the increased hydrophobicity and high energy difference with water result in the lower friction coefficient under boundary lubrication [22]. However, because UHMWPE exhibits a decreased contact angle reinforced by GO and the UHMWPE composites show the enhanced tribological properties, the author believes that increased wettability is beneficial to improve the friction and wear resistance properties under seawater lubrication [27].

As described above, the tribological properties and bearing capacity of polymer materials can be improved by adding different types of fiber or particles under dry sliding or water lubrication conditions. HDPE is widely used as water-lubricated bearings because of its excellent chemical stability and good creep strength. The temperature rise will occur when the bearing undergoes high-speed and heavy-load conditions or in a thermal water environment, such as geothermal water and bio-implant applications, which accelerate the wear and aging of polymer composites [28,29,30]. Inorganic fillers, as a good additive, possess high temperature stability and can effectively enhance the tribological and bearing capacity of composites [31,32,33]. The carbon fiber is an appropriate additive to modify the surface and mechanical properties of the HDPE materials. This study focused on the effects of SCFs and temperature on the surface and tribological properties of HDPE composites under water-lubricated conditions, and aimed to investigate the modification mechanism of filler, as well as influence law and mode of action of temperature on the lubrication and wear of composite. For this purpose, we fabricated the HDPE composite reinforced by SCFs through hot extrusion process. The hardness, surface contact angle, and tribological behaviors were evaluated, and the essential enhancement mechanisms of tribological properties were discussed.

## 2. Materials and Experiment

The twin-screw process is one of the most common methods in polymer composite preparation, and the manufacture quality can be controlled by adjusting the speed and temperatures of the twin-screw extruder.

### 2.1. Materials

HDPE micro-particles with a diameter of 50–150 μm were purchased from WANGDA Plastic company, Shenzhen, China (Figure 1b,d). SCFs with diameter of 7 μm and length of 2–5 mm were supplied from ACP Composites, Livermore, CA, USA (Figure 1a,c). The surface hydroxyl treatment of carbon fibers was conducted with a concentrated nitric acid and concentrated sulfuric acid mixed solution (1:3 by mass) to enhance the interface bonding force between the matrix and fiber [34].

### 2.2. Preparation of HDPE Composites

As shown in Figure 2, the fabrication of CF/HDPE composites was based on hot extrusion forming method [35]. The test composite samples consist of HDPE and CF at different contents. The mixtures containing a certain weight fraction of SCFs (0 wt %, 5 wt %, 10 wt %, 15 wt %, 20 wt %, and 25 wt %) and HDPE were extruded through a twin-screw extruder (WLG10, Xinshuo Precision Machinery, Shanghai, China) with a screw speed of 60 rpm and barrel temperature of 200 °C. After compounding for 10 min, the mixtures were then transferred to a micro-injection molding machine (WZS10G-D from Shanghai Xinshuo Precision Instrument Co., Ltd., Shanghai, China). During the preparation process, the barrel temperature was kept at 200 °C while the mold temperature was set as 60 °C, and the pressure was maintained at 20 MPa for 10 min. The size of prepared composites for the tribological test is Φ 50 × 5 mm.

### 2.3. Characterization

The tribological properties of HDPE composites were evaluated using the HRS-2M pin-on-disc high speed reciprocating tribometer (Lanzhou Zhongke Kaihua Technology Development Co., Ltd., Lanzhou, China). The experiment schematic diagram is shown in Figure 3. The sample was first fixed in the carrier through the compact and the carrier was filled with distilled water acting as lubricant, and then the motor drove the stage to reciprocate on the slide rail and the metal pin (45# steel with a diameter of 4 mm). The tests were carried out with a normal load of 180 *N* and speed of 100 R/min, and the reciprocating sliding length was 8 mm. The water was heated by a self-made heating device to maintain the temperature in preset range (20 ± 1 °C, 40 ± 1 °C, and 60 ± 1 °C).

Contact angle measurements were performed using a dynamic contact angle measurement instrument by sessile drop method at room temperature. For each composite, 10 times measurement were performed and the average values were taken as results. The hardness of HDPE composites was measured by a Vickers hardness instruments (TIME TMVS-1, Sinowon Innovation Metrology Manufacture Limited, Dongguan, China) with an indenter load of 0.98 *N*, each sample repeated 5 times for accuracy.

The microstructure of HDPE microparticles and SCFs was characterized by a field emission scanning electron microscope (SEM, Gemini SEM 500 at 10 kV, Carl Zeiss AG, Oberkochen, German) and digital cameras, respectively. The crystalline phase composition of HDPE composites was detected by an X-ray diffractometer (XRD, D8 Advance A25 at 40 mA and 40 kV, Bruker AG, Karlsruhe, German). The morphologies of worn surface of HDPE composites and pins were characterized by using SEM. The wear track depth and three-dimensional profile of HDPE composites were measured through a laser scanning confocal microscope (Olympus OLS4000, Olympus Corporation, Tokyo, Japan).

## 3. Result and Discussion

The tribological behaviors are comprehensive results of mechanical properties and surface properties of polymer composites. Meanwhile, the working condition and ambient temperature also have an important impact on the lubrication and wear properties of composites.

### 3.1. Effects of SCFs on the Hardness and Wettability of HDPE Composites

The hardness of polymer matrix composites plays an important role in friction and wear behaviors of polymer matrix composites. Figure 4 shows the Vickers hardness of HDPE composites reinforced by different content of SCFs. It can be succinctly observed that the hardness of HDPE composites achieved enhance with the addition of SCFs and reached the maximum when the content of SCFs increased to 25 wt %.

The surface wettability of six different types of HDPE composites is shown in Figure 5. It can be seen that the addition of SCFs could cause different levels of increase in contact angle of the HDPE composites. The major trend is that the contact angle increases with the increases in content of SCFs. However, different from hardness variation trends, the HDPE composites exhibit the maximum contact angle of 95.2° when the content of SCFs is 20 wt %, which exhibits the hydrophobicity; the contact angle decreased to 93.5 while the content of SCFs continued increasing. The SCFs have a lower surface energy compared to the HDPE matrix and, thus, cause an increase in contact angle of HDPE composites [36]; nevertheless, the SCFs are prone to agglomeration when the concentration reaches 25 wt %, which might reduce the specific surface area of HDPE composites. As a result, the contact angle decreased.

The XRD pattern of 20 wt % SCFs filled with HDPE composites is shown in Figure 6. The diffraction peaks at 2θ = 21.4° and 24.8° are assigned to plane of the HDPE matrix. After enhancing by SCFs, the peaks at 2θ = 23.5° are observed, which is assigned to (002) plane of cubic spinel crystal structure of SCFs. The results indicate that the significant modification of crystal structure of HDPE occurred with the addition of SCFs.

### 3.2. Effects of Temperature on Tribological Properties

Figure 7 shows the tribological properties of six types of HDPE composites filled with different content of SCFs under different temperatures. From Figure 7a it can be seen that when under 20 °C, the addition of SCFs can cause the obvious reduction in the friction coefficient. For the pure HDPE, the friction coefficient decreases with running time and reached a stable value after about 30 min, and the composites have a shorter running time when filled with 5 wt % SCFs. From Figure 7b,c, all types of composites showed the more stable friction curve with increasing temperature. The average friction coefficient of six types of HDPE composites under different temperatures is shown in Figure 7d. It can be seen that the friction coefficient decreases with the increases in content of SCFs and reached the minimum value when the content of SCFs increased to 20 wt % at any temperature. After that, the friction coefficient of HDPE composites increases with the SCFs proportion further increasing to 25 wt %. The HDPE composites showed a better lubricity at 60 °C compared to that under 20 °C, and reached the lowest friction coefficient of 0.072 when the ambient temperature increased to 40 °C, which shows an excellent lubrication performance compared with previous studies and similar tendencies with changes in temperature [30,31].

The worn surface of HDPE composites with different content of SCFs at 20 °C are analyzed to evaluate the wear resistance of HDPE composites, and the results are shown in Figure 8. For the neat HDPE, the wear furrows and holes and large cracks were generated on the sliding surface caused by the peeling off of the matrix, indicating a transfer of matrix. The boundary lubrication is the dominant lubrication mode in this study because of the lower sliding velocity (8 mm/s), which was unable to prevent the direct contact between sample and counterpart; the break of the matrix therefore occurred due to its low strength and viscoelastic and the wear mode of neat HDPE was primarily adhesive wear. As shown in Figure 8b, the holes with smaller sizes are observed on the surface of HDPE composites containing 5 wt % SCFs, and the wear furrows were also generated. The HDPE composites possess enhanced hardness and bearing capacity with the addition of SCFs, resulting in the better wear resistance. When the SCFs content increased to 10 wt %, HDPE composites exhibited a similar worn morphology to that reinforced by 5 wt % SCFs from Figure 8c.Theholes are also observed on the worn surface, showing a smoother worn surface, while the fiber pull-out is not obviously observed. As shown in Figure 8d, besides holes, the short fiber is also observed on the wear of HDPE composites when the content of SCFs is 15 wt %. From the macroscopic scale, the high filling content led to the high hardness of HDPE composites, resulting in a reduced contact area between HDPE composites and counterpart; as a result, the friction coefficient decreased. Additionally, from the meso-scale, the contact number between fiber and counterpart increases with increasing SCFs content, which results in a more continuous contact between counterpart and fiber. As a result, the HDPE composites exhibited a more stable friction curve compared to that filled with low content SCFs. Wear mode could also transit to the abrasive wear because of the high hardness and brittle characteristics of SCFs, which causes a decrease in wear loss of HDPE composites. From Figure 8e, many smooth regions can be seen on the worn surface of HDPE composites filled with 20 wt % SCFs, while only broken fibers are observed, indicating a better wear resistance of HDPE composites compared with that filled with 15 wt % SCFs. However, it can be seen from Figure 8f that many cracks were generated on the worn surface of HDPE composites when the content of SCFs further increased to 25 wt %; the main reason can be attributed to the stress concentration around SCFs caused by the agglomeration of SCFs when the volume content of filler is relatively high [11,37].

Figure 9 shows the wear track profiles and three-dimensional morphologies of 20 wt % SCFs filled HDPE composites at 20 °C, 40 °C, and 60 °C, respectively. As shown in Figure 9a, the HDPE composites have a rough wear region at 20 °C. When the water temperature is 20 °C, the matrix of HDPE composites exhibits relatively high hardness and low toughness, thus the wear debris are more likely to fall off from the sliding surface in large sizes, forming a rough wear surface. The plasticity of the matrix increases with the increase in temperature, hence how the relatively smooth wear track profile was generated, which provides an important reason for the decrease in friction coefficients with increasing temperature. The temperature rise is not a positive effect on the wear resistance of the HDPE matrix; the high temperature causes the low strength and bearing capacity of HDPE matrix, so the wear resistance of HDPE composites is relatively poor. The variations significantly disagree with the result of polymer composites under the dry sliding condition [29]. The high temperature generated by friction promotes the formation of transfer film and graphitic-like crystallized structure, so the composites exhibited a reduction in wear loss when above the critical temperature. In this study, the temperature of interface was below 100 °C due to the presence of water, thus phase changes may not occur. However, the temperature has different effects on the mechanical properties of HDPE matrix and carbon fiber. As the polymer material, the larger decline will occur in both elasticity modulus and hardness of HDPE matrix than carbon fiber [38,39]. This phenomenon caused the more severe strain and stress disaccord of matrix and fiber. As a result, the composites showed a decrease in interface bonding force between matrix and fiber, and the debonding and removals of fibers was apt to occur. The increased plasticity caused by high temperature led to severe deformation of matrix—the abundant wear debris is generated in the wear track margin (Figure 9b,c)—indicating the sharply deteriorating wear resistance at high temperature.

### 3.3. High Temperature Wear Properties

In order to further evaluate the high temperature wear properties of HDPE composites, all kinds of HDPE composites were further investigated, the results as shown in Figure 10. It can be seen that the neat HDPE exhibits the worst wear resistance. Nevertheless, the significant improvement is achieved by introducing SCFs to matrix and exhibiting the lowest wear volume until the content of SCFs reached 20 wt %. As the content of SCFs content increased to 25 wt %, the wear track profile HDPE composites exhibits a rough characteristic. The reason is mainly due to the appearance of stress concentration caused by intensified fiber aggregation at the high temperature, the wear debris was removed from the sliding surface in a large size.

### 3.4. Water Lubrication and Wear Mechanism of SCFs

The research in previous parts mainly focuses on the influence factors of friction behavior of HDPE composites in terms of changes in mechanical properties caused by the addition of SCFs. However, these effects do not fully explain the reduction in friction coefficient and wear loss. It is reported that the variation in surface energy of composites may be another important factor to consider [40]. Figure 5 shows an increase in contact angle of HDPE composites when increasing the SCFs content, and the contact angle reached 95.2° when the SCFs content was 20 wt %; namely, the composites exhibit the hydrophobicity. As shown in Figure 11a, the water molecule is more easily adsorbed on the surface of neat HDPE due to its low surface energy; the water absorbing of matrix is thus enhanced and then results in the decrease in the hardness and plasticizer resistance of neat HDPE. As shown in Figure 11b, the hydrophobic HDPE composite and the hydrophilic pin are beneficial in reducing the formation of transfer film. Liquid slipping velocity in a region that is close to the solid surface is increased for liquid contact with hydrophobic or low-surface energy materials, resulting in faster water flow, as well as more efficient heat diffusion [41,42]. These factors make SCFs reinforced HDPE composites more adapt to the water lubrication condition; however, the surface and block properties of HDPE composites changed with the high water temperature. The high temperature results in the low hardness and contact angle. As a result, the bearing capacity, surface flow velocity, and plasticity of HDPE composites is reduced, so the HDPE composites show decreased water lubrication at high temperature.

## 4. Conclusions

In this study, the SCFs were introduced to HDPE to enhance the mechanical and tribological properties of HDPE composites under water lubrication conditions. The SFCs fraction and water temperature on the surface wettability, mechanical, and tribological properties of HDPE composites were investigated and related mechanisms were illustrated. The main conclusions are drawn as follows:(a)The incorporation of SCFs increased the hardness of HDPE composites, and simultaneously improved the contact angle. The hardness increased by 42.9% after adding 25 wt % SCFs, and the contact angle achieved the maximum of 95.2° when the content of SCFs was 20 wt %.(b)The HDPE composites exhibited the reduction in friction coefficient and wear loss compared with neat HDPE, and showed the lowest friction coefficient of 0.076 when the content of SCFs was 20 wt %. The addition of SCFs improved the hardness and bearing capacity of HDPE, resulting in the smaller contact area of composites and counterpart, so wear loss and friction coefficient were reduced. The incorporation of SCFs decreased the surface energy of HDPE composites, which cause the faster water flow near the surface of composites, and the matrix adhesion and heat diffusion decreased.(c)The friction coefficient and wear resistance achieved reduction for each type of HDPE composites at the higher temperature. The enhanced plasticity caused by temperature rise decreases the shear force for the matrix and leads to low friction coefficient. However, the decline in hardness and increase in stress disaccord at high temperatures do not contribute to increased wear resistance. Moreover, the high temperature changes the water flow state in near surface of HDPE composites, which also results in the poor lubrication.

## Figures and Tables

**Figure 1 materials-15-04508-f001:**
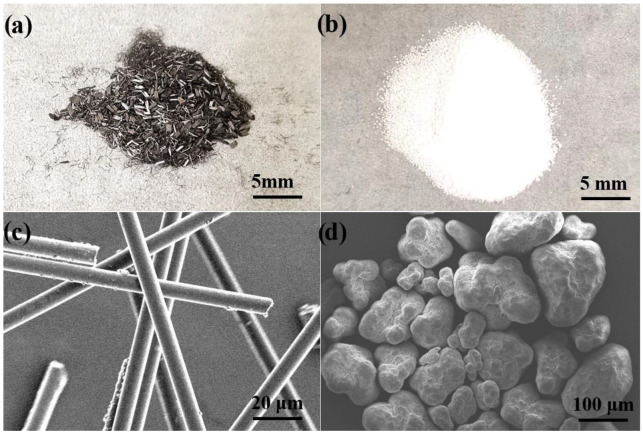
Optical image of (**a**) SCFs and (**b**) HDPE micro-particles; SEM image of (**c**) SCFs, and (**d**) HDPE micro-particles.

**Figure 2 materials-15-04508-f002:**
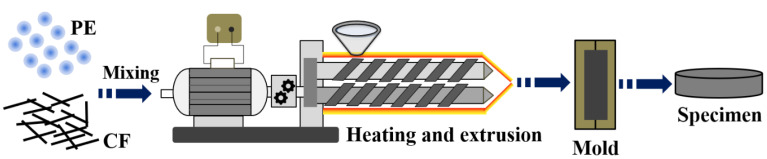
Schematic diagram of the preparation procedure of HDPE composites.

**Figure 3 materials-15-04508-f003:**
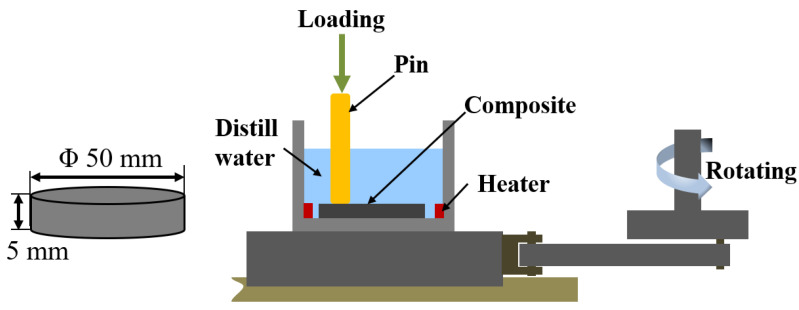
Schematic diagram of tribological characterization.

**Figure 4 materials-15-04508-f004:**
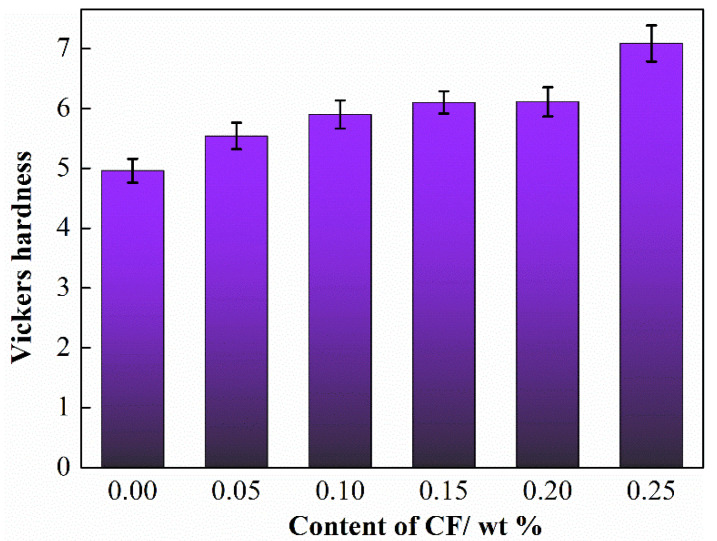
Vickers hardness of HDPE composites with different content of SCFs.

**Figure 5 materials-15-04508-f005:**
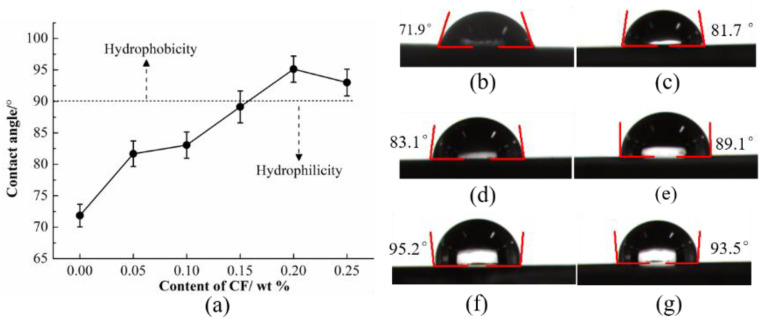
(**a**) Water contact angle of the HDPE composites with different content of SCFs: (**b**) pure HDPE, (**c**) 5 wt % SCFs, (**d**) 10 wt % SCFs, (**e**) 15 wt % SCFs, (**f**) 20 wt % SCFs, and (**g**) 25 wt % SCFs.

**Figure 6 materials-15-04508-f006:**
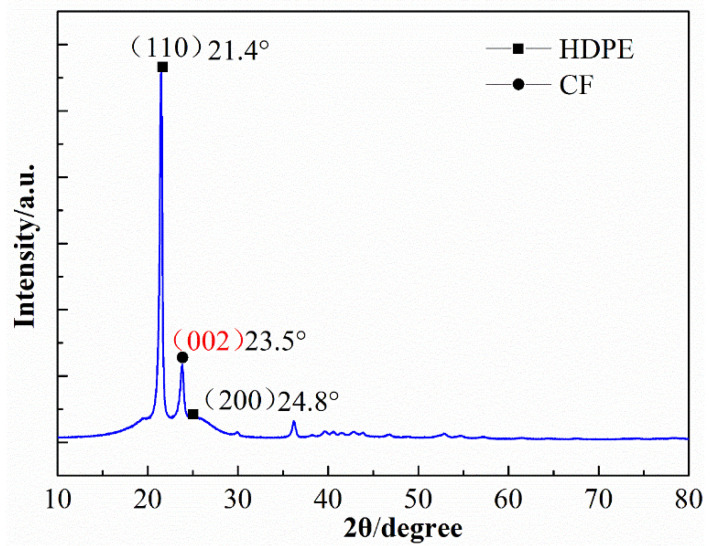
XRD patterns of HDPE composites with 20 wt % content of SCFs.

**Figure 7 materials-15-04508-f007:**
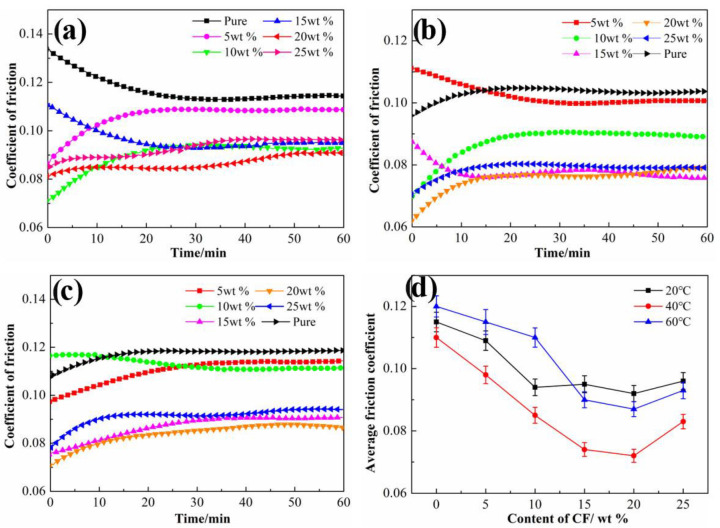
Friction coefficient of HDPE composites with different content of SCFs at (**a**) 20 °C, (**b**) 40 °C, and (**c**) 60 °C, respectively; (**d**) average friction coefficient of HDPE composites with different content of SCFs at different temperature.

**Figure 8 materials-15-04508-f008:**
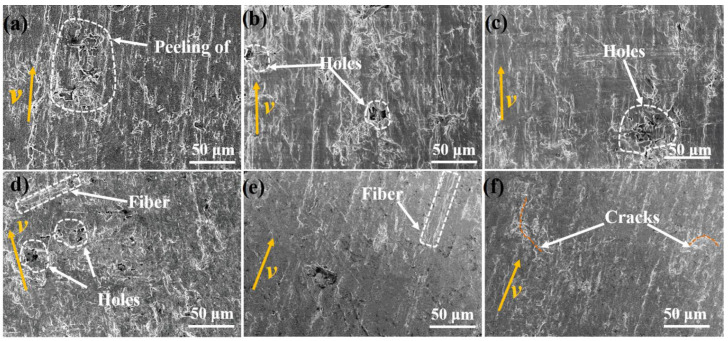
SEM images of worn surface of HDPE composites with different content of SCFs at 20 °C: (**a**) pure HDPE, (**b**) 5 wt % SCFs, (**c**) 10 wt % SCFs, (**d**) 15 wt % SCFs, (**e**) 20 wt % SCFs, and (**f**) 25 wt % SCFs.

**Figure 9 materials-15-04508-f009:**
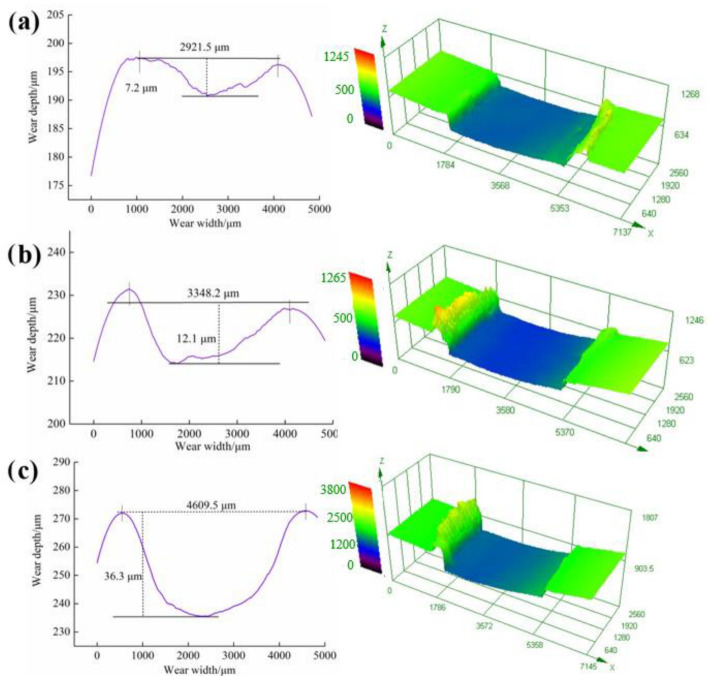
Wear track profile and three-dimensional topography of HDPE composites reinforced by 20 wt % SCFs at (**a**) 20 °C, (**b**) 40 °C, and (**c**) 60 °C, respectively.

**Figure 10 materials-15-04508-f010:**
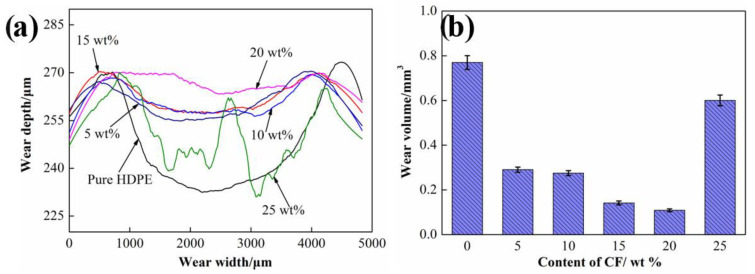
(**a**) Wear track depth of HDPE composites containing different contents of SCFs at 60 °C; (**b**) Wear loss of HDPE composites containing different contents of SCFs at 60 °C.

**Figure 11 materials-15-04508-f011:**
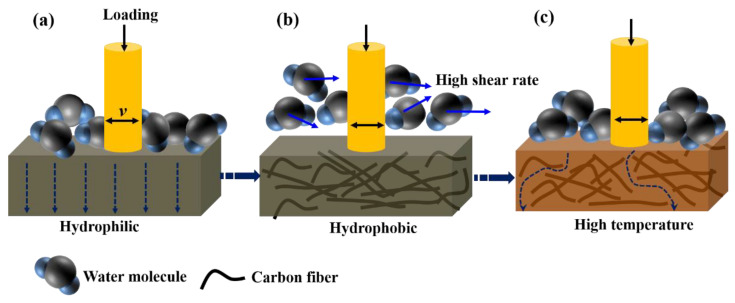
Reinforcing mechanism of SCFs filled HDPE composites at different temperature: (**a**) neat HDPE at 20 °C; (**b**) HDPE composites containing SCFs at 20 °C; (**c**) HDPE composites containing SCFs at 60 °C.

## Data Availability

The data presented in this study are available on request from the corresponding author.

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
