# Peer review of "High-Temperature Tribological Behavior of HDPE Composites Reinforced by Short Carbon Fiber under Water-Lubricated Conditions"

_materials, 2022, doi:10.3390/ma15134508_

Round 1
Reviewer 1 Report
Reviewer Reports:
I recommend major amendments at this level.
General comments:
I reviewed the manuscript entitled “High-temperature tribological behaviour of HDPE composites reinforced by SCF under water lubricated conditions”. The work carried out in the manuscript is interesting and based on fabricated reinforced by SCFs through a hot extrusion process to improve their tribological properties. However, there are several remarks that authors should seriously follow before any possibility of publication. Please carefully check, revise and improve the whole manuscript as there are few syntax/grammatical errors. The service of an expert in the use of English in scientific writings should be sought if necessary. The main novelty in this work must be clearly pointed out. Would you explicitly specify the novelty of your work? What progress against the most recent state-of-the-art similar studies was made? So, the author(s) should be clearer about the uniqueness of the study. The manuscript has a lot of information however there are some lacking connectors. Please remove any multiple references. After that please check the manuscript thoroughly and eliminate all the lumps in the manuscript. This should be done by characterizing each reference individually. This can be done by mentioning 1 or 2 phrases per reference to show how it is different from the others and why it deserves mentioning. This comment is applied all over the paper. Highlights are necessary for this work. Please provide a graphical abstract to provide a visual summary of the main findings of the study. Too many abbreviations are used in the analysis and results. I recommend a nomenclature section for the abbreviations and variables used throughout the passage. The journal's author guidelines and instructions should be followed in preparing the revised version. Some other issues that need to be addressed are:
Detailed comments:
Title: Avoid ALL acronyms in the title.
Abstract:
Please improved the abstract. The abstract should have one sentence per each: context and background, motivation, hypothesis, methods, results, and conclusions. In the abstract, please add an indication of the achievements from your study that are relevant to the journal scope. Please be concise - maximum 1-2 lines. Data should be incorporated into the abstract. Please explain the contributions of the study in the abstract.
Introduction:
The literature review is well presented, however, not strongly linked to the gaps in the research, therefore the novelty of the work is not significant. Please improve the state of the art overview, to clearly show the progress beyond the state of the art. The lack of proper justification creates the wrong impression that the authors are unaware of the recent developments. The aim of the introduction needs to be improved and rewritten. In addition; the introduction should be clearly stated the research questions and targets first. Then answer several questions: Why is the topic important (or why do you study it)? What are the research questions? What has been studied? What are your contributions? Why is it to propose this particular method? Please use relevant recent references by OTHER authors, recent meaning from 2018 - 2022. The relevant reference may be of interest to the author according to below:
https://pubs.acs.org/doi/full/10.1021/acsomega.2c00567
https://www.sciencedirect.com/science/article/abs/pii/S0016236121027289
https://www.sciencedirect.com/science/article/abs/pii/S2352186421001735
Please eliminate the use of redundant words. Eg. In this way, Recently, Respectively, therefore, currently, thus, hence, finally, to do this, first, in order, however, moreover, nowadays, today, consequently, in addition, additionally, furthermore. Please revise all similar cases, as removing these term(s) would not significantly affect the meaning of the sentence. This will keep the manuscript as CONCISE as possible. Please check ALL. Avoid beginning or ending a sentence with one or a few words, they are usually redundant. Kindly revise all.
The aim of the introduction needs to improve.
Materials and Methods:
Please avoid having one heading after another with no discussion in between as in the case of Sections 2 and 2.1. Kindly inspect the entire document for similar instances and revise accordingly.
Results and Discussion:
The authors should perform a comparison between the forecasting results with those of the literature. All the obtained results need to discuss along with the findings of other researchers. when discussing with results, the authors should improve the logic to make it readable. In your discussion section, please link your empirical results with a broader and deeper literature review.
Conclusions:
The conclusion is pretty generic and fails to provide any improvement in the existing knowledge base. The conclusions can still be improved by providing an analysis of where the current work on adsorbents is focused, and what are the remaining gaps in literature where more research should be conducted. How will continued work in this field contribute to or affect the development and adoption of materials? It is recommended to use quantitative reasoning comparing with appropriate benchmarks, especially those stemming from previous work. Please make sure your conclusions section underscores the scientific value-added of your paper, and/or the applicability of your findings/results. Highlight the novelty of your study.
References:
Please double-check the reference section carefully and correct the inconsistency. References style should follow the journal guideline.

Reviewer 2 Report
The authors of the current manuscript investigated the effect of loading an HDPE polymer with various contents of short carbon fiber (SCF) additive on its tribological properties. The manuscript is original and rich in data. However, the following points need to be addressed by the authors:
-The statement of the novelty of this manuscript is not clear and well-written. The use of carbon fiber additives to enhance the morphology, mechanical, and thermal expansion performance of HDPE base material has been reported previously in the literature (see for example, High Density Polyethylene Composites Reinforced with Hybrid Inorganic Fillers: Morphology, Mechanical and Thermal Expansion Performance. Materials (2013), 6, 4122-4138. 10.3390/ma6094122). Can the authors comment on this?
-I suggest for the authors conduct the FTIR analysis on the neat HDPE polymer as well as the polymer formulations that are mixed with different contents of the SCF additives. The comparison of these spectra can provide valuable information on the occurrence of any possible chemical modification on the parent polymer network due to the additive-loading step, especially at high temperatures.
-Regarding the contact angle measurement, I don’t think the insignificant change in the contact angle value from 95.2 to 93.5 as a result of increasing the amount of SCF additive is enough to correlate it with any property. Can you support this result with more references from the literature?
-The English of the manuscript requires major enhancement. See for example lines 176-177, 187-188, and many others.
Reviewer 3 Report
The manuscript tries to describe the fabrication of high-density polyethylene composites reinforced by varying short carbon fibers using twin-screw extrusion and injection molding techniques. The authors studied various parameters such as hardness and surface wettability of the fabricated material under different temperatures. The title and abstract are appropriate for the content of the text. In general, the manuscript is well constructed, the experiments were well conducted, and the analysis was well performed.
Minor comments
· The acronym “PEI” in line 36, “PI” in line 39, “PEEK” in line 53, and “UHMWPE” in line 54 are not defined.
· In line 40, replace the word “mainly” with “main”.
· The sentence in line 82 is awkward. The authors need to clarify what is fabricated in this study.
· From line 245 onward, the acronym “HDPE” is changed to “DHPE”. Are they two different materials? If not, then it should be consistent throughout the paper.
· In line 144, the authors stated the maximum hardness is obtained when the SCFs content is increased to 25 wt%. Does this mean the hardness decreases when the SCFs content is more than 25 wt%? If so, the authors need to show/include the result in Fig.4.
Reviewer 4 Report
The authors present the results of a study of HDPE/SCF composites for tribological applications.
The authors melt-compound HDPE and SCF in a twin-extruder and produce specimens for testing. Testing includes measurement of coefficient of friction, XRD, contact angle, hardness and surface microscopy.
Unfortunatelly, very little attention is placed on the sub-surface microstructure of the molded specimens, the orientation of the fibers, and the possible fiber pull-out during wear testing.
The method of fabrication of the test specimens is not clear; is a master-batch produced in the twin-screw extruder and then fed into the injection molding machine? In what form? pellets? what, if any, is the fiber attrition during compounding ? What is the possibility of the formation of (poorly wetted?) fiber strands during compounding, which might manifest in molded specimens of uneven microstructure? These are important issues that should have been addressed.
In spite of the extensive amount of work involved in this study, there is no new knowledge (or methods, or data interpretation) in regards to the properties and performance of such composites for tribological applications. The study could be a solid technical report, but I cant see it as a research paper, as it does not advance the state-of-the-art in this area. Therefore, the recommendation is to not accept.
PS: In some instances HDPE is mis-spelled as DHPE
Round 2
Reviewer 1 Report
Reviewer 2:
I have reviewed the revised manuscript entitled" High temperature tribological behavior of HDPE composites reinforced by SCF under water lubricated conditions". The work is interesting and it falls within the scope of the journal. Moreover, the authors adequately answered the queries of the reviewer. The paper has been improved and can be accepted. I do not have further comments.
Reviewer 2 Report
The manuscript looks much better now and I recommend it for an immediate publication.
Reviewer 4 Report
The authors have addressed the issues raised in the 1st review, and also provided signifficant supporting information. I therefore believe the revised manuscrip[t to be suitable fr publication in "Materials".